# Molecular Analysis of the Official Algerian Olive Collection Highlighted a Hotspot of Biodiversity in the Central Mediterranean Basin

**DOI:** 10.3390/genes11030303

**Published:** 2020-03-13

**Authors:** Benalia Haddad, Alessandro Silvestre Gristina, Francesco Mercati, Abd Elkader Saadi, Nassima Aiter, Adriana Martorana, Abdoallah Sharaf, Francesco Carimi

**Affiliations:** 1Département de Productions Végétales, Laboratoire Amélioration Intégrative Des Productions Végétales (AIPV, C2711100), Ecole Nationale Supérieure Agronomique (ENSA), Hassan Badi, El Harrach, Algiers 16000, Algeria; b.haddad@ensa.dz; 2Institute of Biosciences and BioResources, National Research Council (CNR), Research Division of Palermo, Corso Calatafimi 414, 90129 Palermo, Italy; francesco.mercati@ibbr.cnr.it (F.M.); adrianamartorana1982@gmail.com (A.M.); abdoallah.sharaf@agr.asu.edu.eg (A.S.); francesco.carimi@ibbr.cnr.it (F.C.); 3University Hassiba Benbouali, Faculty of Science of Nature and Life, Plant Biotechnology Laboratory, BP 151, Chlef 02000, Algeria; a.saadi@univ-chlef.dz; 4Université Saad Dahleb-Blida 1, Faculté des Sciences de la Nature et de la Vie, Laboratoire de Biotechnologie des Productions Végétales, Département de Biotechnologies, Blida 09000, Algeria; 5Laboratoire de culture in vitro, Département central, Institut Technique de l’Arboriculture Fruitière et de la Vigne, ITAFV, Algiers 16000, Algeria; 6Institute of Molecular Biology of Plants, Biology Centre, CAS, Branišovská 31, 37005 České Budějovice, Czech Republic

**Keywords:** *Olea europaea* L., olive, cpSSR, nuSSR, genetic diversity, population structure, Mediterranean Region

## Abstract

Genetic diversity and population structure studies of local olive germplasm are important to safeguard biodiversity, for genetic resources management and to improve the knowledge on the distribution and evolution patterns of this species. In the present study Algerian olive germplasm was characterized using 16 nuclear (nuSSR) and six chloroplast (cpSSR) microsatellites. Algerian varieties, collected from the National Olive Germplasm Repository (ITAFV), 10 of which had never been genotyped before, were analyzed. Our results highlighted the presence of an exclusive genetic core represented by 13 cultivars located in a mountainous area in the North-East of Algeria, named Little Kabylie. Comparison with published datasets, representative of the Mediterranean genetic background, revealed that the most Algerian varieties showed affinity with Central and Eastern Mediterranean cultivars. Interestingly, cpSSR phylogenetic analysis supported results from nuSSRs, highlighting similarities between Algerian germplasm and wild olives from Greece, Italy, Spain and Morocco. This study sheds light on the genetic relationship of Algerian and Mediterranean olive germplasm suggesting possible events of secondary domestication and/or crossing and hybridization across the Mediterranean area. Our findings revealed a distinctive genetic background for cultivars from Little Kabylie and support the increasing awareness that North Africa represents a hotspot of diversity for crop varieties and crop wild relative species.

## 1. Introduction

Olive (*Olea europaea* L.) is one of the most important fruit species of the Mediterranean region [1]. Ninety-eight percent of olive trees of the world are cultivated in this region [2], providing over 90% of World production [3]. Olive fruits and olive oil are central in the Mediterranean diet and symbols of the Mediterranean culture. It is commonly believed that olive domestication occurred in the Near East approximately 6000 years ago [4]. Phoenicians, Greeks and Romans later spread olive cultivation to the western Mediterranean region [5,6,7,8]. The hypothesis of a human-mediated diffusion of the olive tree from the eastern to western Mediterranean basin is supported by recent genetic studies [9], demonstrating that as many as 90% of current cultivars are characterized by the same chloroplast haplotype lineage [4,10]. Therefore, the spreading of the olive culture throughout the Mediterranean Basin by human migrations and commercial exchanges has played a key role in determining the pattern of olive germplasm [11,12]. The cultivated olive germplasm shows a high degree of diversity, with about 1250 recognized cultivars [13]. Olive cultivation in Algeria dates back to antiquity, and it has maintained great socio-economic importance until present days [14] and is mostly present along the Mediterranean coast. In this area, the mountainous region of Kabylie, geographically divided in two districts by the river Soummam, great Kabylie to the West, and little Kabylie to the East, can be considered an important reserve of local olive germplasm [14]. The olive sector is considered strategic for the Algerian economy and for this reason the Algerian Ministry of Agriculture and Rural Development recently set a strategy for the expansion of olive tree cultivation in different regions, aiming to reach one million hectares by 2019, using the local genetic resources. Therefore, the identification and characterization of local germplasm is a key step for future breeding programs, cultivar selection for new plantations and to preserve Algerian olive biodiversity from the risk of genetic erosion due to introduction of foreign cultivars. Despite that Hauville [15] reported 150 varieties in Algeria, according to the Algerian Ministry of Agriculture and Rural Development, 36 main varieties are officially recognized and cultivated in the experimental field of the Institut Technique de l’Arboriculture Fruitiere et de la Vigne (ITAFV, Takarietz, Bejaia). Previous studies on Algerian olive germplasm focused mainly on the genetic characterization of a subset of local cultivars [16,17], their population structure and their genetic relationship with wild olive trees [14]. In his study of the World Olive Germplasm Bank (WOGB) of Marrakech, Haouane et al. [18] analyzed some Algerian varieties with nuclear and chloroplast microsatellites (nuSSRs and cpSSRs), but only their cpSSRs profiles are publicly available.

The present study is the first genetic characterization of the official Algerian National Olive Germplasm Repository, using both nuSSRs and cpSSRs. Among the available molecular markers, nuSSRs were chosen for their highly reproducible and informative co-dominant and multi-allelic nature, which allowed to evaluate the genetic diversity in several plant crops, such as maize [19], rice [20], common bean [21], wheat [22], tomato [23,24], grape [25,26,27,28] and olive [29,30,31,32,33]. CpSSRs are maternally inherited in angiosperms, and they have been informative in unravelling the phylogenetic pattern in olive germplasm [4,34] but also in other crops such as grape [35,36].

In order to provide new insights in the origin and diffusion of olive cultivars around the Mediterranean basin, we analyzed the Algerian varieties with 16 nuSSR and six cpSSR markers and compared our results with the widest published datasets available, which includes olive varieties representative of Mediterranean Basin crop’s biodiversity. The aims of this research were: (i) to evaluate the genetic diversity of the main Algerian olive varieties; (ii) to assess, for the first time, the genetic relationships between this germplasm and olive accessions from public datasets; (iii) to provide useful knowledge for future cultivation expansion and breeding programs.

## 2. Materials and Methods 

### 2.1. Plant Material and Sampling

A total of 34 Algerian varieties from the ITAFV national olive germplasm collection (Table 1) were sampled for the genetic characterization. The ITAFV experimental field was created between 1947 and 1954, covering 0.95 ha. It is located 30 km off Bejaia Takarietz (latitude 36.24, longitude 6.57 and altitude 63.30), in the coastal area of the Sidi Aich district (Figure 1), an area with an arboriculture vocation, characterized by a Mediterranean climate [37]. Information on Algerian varieties including its Arabic name, meaning, synonyms, putative origin, diffusion and use are reported in Appendix A, the catalogue illustrating the main features of each variety is presented in Appendix A.

### 2.2. Molecular Analyses (nuSSRs and cpSSRs)

Total genomic DNA was extracted from 0.1 g of dry leaves following the Doyle and Doyle [38] CTAB (cetyl-trimethylammonium bromide) method. The extract was treated with DNase-free RNase (Roche Diagnostics, Mannheim, Germany) and the quality and concentration were checked by a NanoDrop Spectrophotometer (Thermo Scientific—Waltham, MA, USA).

Algerian varieties were analyzed by 16 nuSSRs available in current literature [28,29,30,31] (Appendix A). The haplotype of each variety was evaluated by using six cpSSRs [34] (Appendix A). Multiplexed amplification reactions were performed in 15 μL final volume reaction mixture as described by Garfì et al. [39]. The amplification products were solved on ABI PRISM 310 Genetic Analyzer (Applied Biosystems by Life Technologies, Foster City, CA, USA) and the alleles were sized by GENEMAPPER 4.0 (Applied Biosystems by Life Technologies). 

Many articles analyzed large datasets of wild and cultivated olive nuSSR profiles, but unfortunately, only a few of them provide their genetic profiles. We compared our nuSSRs profiles with the largest published dataset available from the WOGB of Cordoba [40], using a common subset of seven SSRs (Appendix A). Normalization among datasets was achieved using the common variety Chemlal de Kabylie, present in our dataset with the synonym Chemlal. For all the analyses, WOGB profiles were grouped geographically as follow: Spain and Portugal (Iberian Peninsula—IB); France (FRA), Italy (ITA), Morocco and Tunisia (Maghreb—MAG); Croatia, Albania and Greece (Balcanic Peninsula—BAL), Turkey and Cyprus (Turkey—TUR); Iran, Israel, Lebanon, Syria and Egypt (East Mediterranean—East-M). Due to the reduced number of SSRs, WOGB dataset was reduced to 351 accessions to consider only unique genetic profiles, and some Algerian varieties were grouped in single genetic profiles for the Structure analyses because with seven SSRs they were not able to differentiate, namely: Aguenau, including Agrarez and Hamra, and Aimel, including Aberkane. cpSSRs profiles were also compared with the available published dataset [4,34]. 

### 2.3. Data Analysis

For each microsatellite we estimated the principal genetic parameters, i.e., number of alleles (Na), expected (He) and observed (Ho) heterozygosity and Polymorphic Information Content (PIC) by using PowerMarker [41], Haplotype analysis software version 1.05 [42] and FreeNA [43] software.

To identify the number of genetic groups in the Algerian germplasm, cluster analysis was carried out for nuSSR and cpSSR separately according to the UPGMA (Unweighted Pair-Group Method with Arithmetical Averages) algorithm and two phylogenetic trees were generated using the R package Adegenet [44]. The levels of support for the nodes were estimated by bootstrap analysis (1000 replicates). The number of genetic groups within the Algerian collection alone and among Algerian and other Mediterranean and Near East cultivars, was inferred by means of Principal Coordinates Analysis (PCoA) in GenAlEx v.6.51b2 [45] and of Bayesian analysis in STRUCTURE [46]. The most likely number of genetic groups (*K*) in STRUCTURE was calculated following Evanno et al. [47]. Twenty independent runs (100,000 burn-in, 1,000,000 Marchov Chain Monte Carlo) for each *K* were carried out using the admixture model with correlated marker frequency and default parameters. The runs were averaged using CLUMPP (CLUster Matching and Permutation Program) [48] and the histograms were shown using DISTRUCT program [49]. Individuals with ancestry value < 0.65 were considered mosaics (Appendix A), while those with higher values were assigned to the corresponding cluster. Using nuSSR profiles, main genetic parameters, including pairwise Gst values [50], among Mediterranean populations were calculated in GenAlEx v.6.51b2 [45]. For Gst values, the significance of the differentiation between pairs of selected populations was tested by permutation procedures (9999 replicates).

For the comparison with the WOGB dataset [40], three STRUCTURE analyses were performed. The first analysis tested whether the number of Mediterranean olive genetic clusters (West, Central and Eastern Mediterranean) changed by the reduction of the SSRs marker panel. The second analysis identified the ancestry of the Algerian germplasm. Finally, a hierarchical analysis [51] following the procedure described above, was carried out only with samples belonging to Central and Eastern Mediterranean pool, showing an ancestry value higher than 0.80 (Appendix A).

## 3. Results

### 3.1. Genetic Diversity Assessed by cp SSRs and nuSSRs

The six cpSSR markers showed a total of 12 alleles, with an average of two alleles per locus and major allele frequency values ranging from 0.647 to 1.000 (Appendix A). Five out of six cpSSR used were polymorphic, while trnT-L-polyT locus was monomorphic in the analyzed collection (Appendix A). The phylogenetic tree obtained through cpSSRs (Figure 2) highlighted three chlorotype groups, corresponding to the wild and cultivated lineages identified in the Mediterranean (Table 2) [4].

NuSSRs markers amplified a total of 140 alleles, ranging from five to 11 for EMO90 and DCA07, respectively, with an average of 7.2 alleles per marker (Appendix A), which is in agreement with previous studies [14,52]. The average PIC value for nuSSRs (0.659) indicates that the analyzed markers are highly informative and useful for variety screening. Among nuSSR loci, 11 (69%) showed high polymorphism with PIC values exceeding 0.6 (Appendix A). In agreement with the PIC value, the average He value was 0.716, while the Ho value was greater than 0.500 for 10 loci (Appendix A), underlining a remarkable rate of heterozygosity among the studied cultivars. 

The UPGMA phylogenetic tree based on nuSSRs underlined a main genetic group (A) of 21 varieties, 13 of which belonged to Little Kabylie (LK), split in four subclusters. A second group (B) of 13 cultivars separated in two subclusters (Figure 3a). Within group A, subcluster A1 included only cultivars native to LK; A2 consisted of two varieties from LK plus Sigoise and Neb Djemel, considered native to the Mascara Plain (West from LK) and Cherchar (South-East from LK), respectively; A3 showed three varieties from LK, Hamra from the nearby coastal area of Jijel, and Mekki from the Aurès mountain region (close to the Sahara desert); A4 included Chemlal, an important variety that covers 30% of the Algerian olive orchard, and three cultivars with local distribution, i.e., Bouricha, Grosse du Hamma and Rougette de Mitidja. In the B group, the two subclusters accounted six varieties each, with three cultivars considered native to Kabylie, i.e., Akerma, Bouchouk Soummam and Tabelout. The last remaining cultivar, considered native to Kabylie, Aghchren d’el Osseur, clustered alone as an outgroup.

STRUCTURE analysis clearly assigned 30 varieties (ancestry value > 0.65) to one of the six identified genetic groups (Figure 3b, Appendix A), while the remaining four cultivars (Aeleh, Blanquette de Guelma, Bouricha, Neb Djemel) showed a mosaic genetic pattern. The six genetic groups were in agreement with phylogenetic analysis, with group K1, K2, K4 and K5 included in cluster A, counting the 13 varieties from LK, while K3 and K6 belonged to group B. In particular, K4 and K5 exactly corresponded to group A1 and A3, respectively; K1 included the variety belonging to A2, plus Grosse du Hamma; in K2, two varieties of A4 (Chemlal and Rougette de Mitidja) plus the cultivar Aghchren d’el Osseur were included. Finally, K3 and K6 account respectively for three and seven varieties, but they did not correspond to the subclusters B. 

We further analyzed the nuclear genetic profiles by PCoA (Appendix A). The resulting pattern reflected the genetic structure identified by UPGMA and STRUCTURE: the first axis separated most of the LK varieties with some other cultivars from group A. The second axis separated a group of five varieties, corresponding to A3. 

### 3.2. Relationship among Algerian and Mediterranean and Near East Germplasm 

In order to frame the genetic relationships of the Algerian cultivars within the three main Mediterranean lineages, we compared our profiles with a large dataset of cultivated accessions across the Mediterranean Basin and Near East [40] using hierarchical and Bayesian clustering by mean of a common set of seven nuSSRs. The genetic parameters for each population are shown in Table 3. We observed high values of genetic diversity for each population (ranging from 0.635—FRA—to 0.746—EAST-M, mean 0.698), and a mean of 0.772 for observed heterozygosity, with the Algerian cultivars that showed the lowest value (0.696). The inbreeding coefficient was negative for all populations, but can be considered in equilibrium. 

The pairwise Nei’s genetic distances and relative Gst values (Table 4) indicated that the Algerian and Iberian germplasm were significantly (*p* < 0.01) the most unrelated, followed by Algeria vs. France and Algeria vs. Turkey. On the contrary, the most related cultivars were those from Turkey vs. East-Mediterranean, Italy vs. Balkan, Balkan vs. Turkey and Iberian vs. Maghreb. The PCoA analysis was able to separate a wide number of the Western varieties from the Central-Eastern group. The Algerian samples showed a bimodal distribution with a group of varieties located in the center of the graph with the Turkish and Near Eastern cultivars, and a second group clustering mainly with the central Mediterranean varieties from Italy and Balkan Peninsula (Figure 4a). The UPGMA phylogenetic tree differentiated Algerian germplasm in two subclusters (Appendix A), one with Western affinity consisting of Iberian Peninsula and Maghreb accessions and the other intermingled with mainly East Mediterranean cultivars and with a minor contribution of Iberian and Balkan varieties.

STRUCTURE analysis of the whole dataset revealed that the most likely number of clusters of Mediterranean varieties (*k* = 2) was in agreement with previous results [40]. In particular, as reported in other studies [4,9,53], two main pools were identified (Figure 4b), the first (A—orange) accounting for the most part of Western Mediterranean varieties, and the second (B—light blue) including mainly Central and Eastern Mediterranean cultivars. Algerian varieties clustered mostly with cluster B (59%), 9% was grouped in cluster A and the remaining cultivars (32%) were mosaics between the two groups (Appendix A). Interestingly, all the cultivars belonging to cluster B were from LK, i.e., Aghenfas, Tefah and Bouchouk Guergour, the latter already differentiated by UPGMA analysis.

Finally, to have a higher resolution of group B, the more heterogeneous pool, which includes samples from all population studied, we ran a second round of STRUCTURE analysis using only the samples closely associated to it (ancestry >0.8). In total, 18 Algerian cultivars and 126 accessions from the entire Mediterranean region were investigated. The analysis allowed to identify four subclusters, including 84 varieties with strong association (ancestry value >0.65), while the remainders 60 cultivars were mosaics (Figure 4c, Appendix A). Subcluster B1 included an assorted group consisting of samples from the entire Mediterranean Basin dominated by Central-Western Mediterranean cultivars including 100% of French accessions, 77% Italian, 57% Balkan and 44% Iberian, with a minor contribution of Turkish (31%), Maghreb (20%), East-Mediterranean (15%) and Algerian (6%). Subcluster B2 was well represented by North African cultivars accounting for 50% of the Algerian germplasm and 60% of Maghreb accessions, with a minor contribution of Turkey (15%), East-Mediterranean (15%), Balkan (13%) and Italy (8%). Subcluster B3 included mainly Eastern and Central Mediterranean varieties, accounting for the 41% of East-Mediterranean accessions, 33% Algerian, 23% Turkish, 15% Italian, 20% Maghreb and 4% Balkan. Subcluster B4 included almost exclusively Western and Eastern cultivars, accounting for 56% of Iberian varieties, 31% Turkish, 28% East Mediterranean and 26% Balkan, with a smaller contribution of central Mediterranean accessions (11% of Algerian accessions). 

## 4. Discussion

For thousands of years, olive cultivation has been central in the culture and economy of many Mediterranean and Middle Eastern regions. The ancient civilizations that thrived in this wide geographical area selected and diffused countless varieties across the different countries facing the Mediterranean Sea. Due to these complex historical events, an endless debate arose among scholars about olive domestication, and in particular whether there has been a single or multiple independent domestication events [54,55]. 

In the last decades, the development of molecular markers such as nuclear and chloroplast SSRs have made it possible to investigate the genetic fingerprint of cultivated and wild olives and disentangle the clues left by migrations and crossing among varieties across the entire olive distribution area. Despite many recent studies provided nuclear and chloroplast SSRs genetic profiles for hundreds of wild and cultivated olive accessions, the germplasm from Central and Southern Mediterranean regions, especially from the Maghreb area, is highly underrepresented. Here, for the first time, we characterized the official Algerian collection of olive varieties from ITAFV by mean of both nuSSRs and cpSSRs. Our results filled the gaps left by previous studies [14,16,17] as we provided the nuSSR genetic profiles for 34 out of 36 official Algerian varieties, 10 of which have never been described (Aimel, Bouchouk Guergour, Bouchouk Lafayette, Boukaila, Grosse du Hamma, Hamra, Longue de Miliana, Mekki, Neb Djemel, Ronde de Miliana) and, for the first time, cpSSR profiles for seven varieties (Aeleh, Aghenfas, Bouchouk Guergour, Boughenfous, Bouichret, Tabelout, Tefah). Overall, high genetic diversity was observed, in agreement with the range obtained in previous studies [5,40,56,57], indicating that the Algerian olive germplasm collection represents an important genetic reservoir for the species. Compared with previous studies on Algerian germplasm [14,17], we found a lower number of alleles but a remarkably higher observed and expected heterozygosity. These discrepancies are probably due to the different panel of varieties analyzed and the presence of wild germplasm in previous studies, which contained private alleles [14,17]. The 16 nuSSRs used here allowed us to discriminate all the Algerian cultivars and were powerful enough to resolve putative cases of synonymy. For example, we found that Agrarez and Azeradj had distinct profiles at eight loci and should not be considered as synonyms as previously suggested [14]. Most of the Algerian varieties (70%) belonged to the Mediterranean/Saharan Africa chlorotype olive lineage E1, widely represented in the cultivated and wild forms in the whole Mediterranean Basin. This cluster included two subclusters, CE1-CL1 and CE2 [34]. Subluster CE1-CL1 consisted of 13 varieties from LK region, Hamra from the nearby coastal area of Jijel, Longue de Miliana and Sigoise from the Central-West regions, Mekki, Ferkani and Souidi from Aurès. Subcluster CE2 accounted for one variety from LK (Boughenfous), and three varieties from the territory of Costantine (Grosse du Hamma), Guelma, (Blanquette de Guelma) and Aurès (Aeleh), respectively. Seven varieties, two from LK (Aghchren d’el Ousseur and Tabelout) and five from other regions (Abani, Limli, Neb Djemel, Ronde de Miliana, Rougette de Mitidja) grouped in the Central-Western Mediterranean lineage E2. Interestingly, this lineage is represented by wild olive mainly from Italy and Greece with a minor contribution from Spain and Morocco, and by few cultivars (*n* = 13) from different central Mediterranean regions (Corsica, France, Greece, Italy, Morocco, Sardinia, Spain and Tunisia) [58]. Chemlal, Akerma and Boukaila clustered in the other less common Western Mediterranean lineage E3, mainly found in wild olive from Spain and Morocco, and in cultivars from Maghreb except for three varieties from Corsica, France and Spain (i.e., Antonina, Olivière and Farga), respectively [58]. Our results mostly confirmed the chlorotypes identified in previous studies (*n* = 18), but highlighted some divergence (Table 2); in particular, we found that nine varieties chlorotypes were assigned differently compared to Besnard et al. [4], six when comparing Besnard et al. [4] and Haouane et al. [18], and two when comparing the three datasets. These results could be due to *i)* possible mislabeling errors in the WOGB collection of Marrakech; *ii)* errors in the published datasets, at least for the same six varieties coming from the above mentioned collection that showed different chlorotypes between the dataset of Besnard et al. [4] and Haouane et al. [18]; *iii)* different clones of the same varieties. The different clustering methods adopted in our study highlighted a clear genetic group mainly consisting of 13 LK cultivars, except for a few varieties (four) from other regions (emigrants). Conversely, a few cultivars (four) from LK clustered in other genetic groups (in-migrants). We can formulate some speculative hypotheses to explain these few exceptions to the general geographic and genetic division between LK region and other parts of Algeria. Emigrant varieties might share a wide genetic background with the LK group because they were selected in this region, but later, for some ecological/historical/agronomic reasons, their cultivation disappeared from the LK area and the knowledge of the original native region was lost. This could be the case of Mekki, Neb Djemel and Hamra, today cultivated only in the driest mountain area of the Aurès or Kenchela or in the coastal area, respectively, or in contrast diffuse throughout Algeria (Sigoise). It has been documented that the historical distribution of Mekki was connected to Phoenician, Greek and Roman dominations [37]. This variety share the chloroplast haplotype dominant in LK varieties, suggesting a common origin, thus, it is plausible that it was once distributed in this area and that it later disappeared in LK remaining confined in a restricted area near the old Roman town of Timgad. For in-migrant cultivars, we can speculate that the knowledge of the original native region was lost: these varieties could have been selected and genetically improved in LK by crossing with germplasm imported from other regions of Algeria/North Africa, thus explaining the genetic divergence from other LK varieties. In particular, nuSSR results were supported by cpSSR analysis indicating that *in-migrant* cultivars had different haplotypes as compared to other LK accessions. In a wider perspective, the combined use of nuSSR and cpSSR, that are differently affected by evolutionary processes (i.e., selection, mutation, recombination etc.), allowed us to investigate the genetic relationship of Algerian varieties with cultivars from other Mediterranean countries and shed light on the geographic origin of Algerian germplasm and relative patterns of crossing/migration across the Mediterranean Region. Our results are compatible with two different hypotheses: (i) local domestication from oleaster and Laperrine’s olive; (ii) local selection by crossing of cultivars imported from other Mediterranean region with local germplasm. The gene flow between Algeria and the rest of the Mediterranean area has been probably limited, suggesting that development of new cultivars possibly proceeded through crossing of few imported varieties with local germplasm, namely oleaster and Laperrine’s olive, as testified by cpSSRs pattern of genetic diversity. We found that the highest proportion of Algerian varieties shared the same haplotype with the majority of Mediterranean cultivars (E1) and the Laperrine’s olive. Interestingly, 30% of Algerian varieties belonged to the other two lineages E2 and E3, unravelling the contribution of the Western Mediterranean olive lineage to the origin of North African cultivars. This result provides new evidence on the role of Algeria as possible and important secondary domestication/selection center, considering that only 4.9% and 4.4% of the Mediterranean cultivars belong to E2 and E3 lineages, respectively [4]. In particular, we can speculate that few varieties with chlorotype E2 probably represent locally domesticated cultivars or cultivars imported from a central Mediterranean region such as Italy, e.g., during Roman domination, as this haplotype is only found in few cultivated varieties from Central Mediterranean region and Maghreb but it is common in wild olive from Italy and Greece. In addition, the three cultivars belonging to lineage E3 represent the most likely candidates for secondary domestication events in this area, given that E3 haplotype is rare and found in wild oleaster from Spain and Morocco and cultivated varieties from Morocco (*n* = 10), France (*n* = 2) and in one variety from Italy and Spain each. 

Finally, regardless of the true origin of the Algerian germplasm, LK varieties can be considered an exclusive genetic core, selected and developed during the different historical periods by the civilizations that thrived in this area from Phoenicians to Arabs until the present. Genetic differentiation parameters and results of the hierarchical STRUCTURE provided evidence that Algerian varieties are more genetically related to Central-Eastern Mediterranean cultivars than to the West. In particular, the second round of STRUCTURE highlighted four main subclusters among the group of Eastern varieties. The Algerian germplasm grouped mostly in two subclusters, both with Central and Eastern affinity. Subcluster B1 was particularly interesting because it was dominant in Algeria and Maghreb, suggesting gene flow between Near East and North Africa. We speculate that the two STRUCTURE clusters could correspond to bottleneck events due to the arrival of different civilization, i.e., Phoenician and Romans. According to the chlorotype lineages identified, 67% of Algerian varieties showed affinity to the Eastern Mediterranean germplasm and 33% to the West. Nuclear DNA confirmed this pattern, with 80% of varieties showing affinity to the Eastern cluster and 20% to the Western cluster. These results might depend on recurrent reticulation events during the diffusion of olive culture [34] and reflect the predominant role of Eastern germplasm in the development of olive cultivars around the whole Mediterranean Basin [54,59]. However, our study revealed an important contribution of Central-Western Mediterranean germplasm in the development of olive varieties, supporting the hypothesis of the existence of an independent domestication center in the Central Mediterranean area [9,55]. This hypothesis is supported by the fact that the wild Laperrine’s olive tree share its haplotype (E1) with most of the world’s olive varieties, including 67% of the Algerian varieties, whereas the remaining 33% share their haplotypes (E2 and E3) with wild olives from Spain, Morocco, Italy and Greece. In particular, given that North African germplasm is highly underrepresented in current literature, the role of this area in the history of olive domestication and cultivar development should be reconsidered to evaluate its real contribution.

Our present data do not allow discriminating between the two different hypotheses, namely the occurrence of a secondary domestication event or introgression of imported cultivars with local germplasm, including natural populations (oleaster and Laperrine’s olive). We can hypothesize that before foreign civilizations arrived in Algeria, wild olive tree populations consisted of two taxa, oleaster (*Olea europaea* subsp. *sylvestris*) and Laperrine’s olive (*O. europaea* subsp. *laperrinei*), that were already exploited by local human population, laying the foundations for the development of olive cultivation. Subsequently, phoenicians introduced Eastern Mediterranean olive trees to North Africa, triggering gene flow with the local germplasm (oleaster and Laperrine’s olive) and with cultivated olive coming from other Mediterranean areas. Local people and settlers from abroad (e.g., Romans, Arabs) eventually selected the cultivars better suited for the cultivation in the different environmental conditions of each Algerian region, thus originating the distribution pattern that we observe today. 

## 5. Conclusions

Genetic studies of local olive varieties from different Mediterranean areas, in particular from North Africa, central Mediterranean area and Near East, can help to clarify the pathways of domestication and diffusion of this species along the history of civilizations. Due to its central position in the Mediterranean Basin, Algeria has played an important role for civilizations crossing the Mediterranean Basin, especially for Phoenicians, Romans and Arabs. To the best of our knowledge, this is the first study analyzing and characterizing the official Algerian olive germplasm collection by means of nuclear and chloroplast SSRs, comparing the results with available published datasets. An exclusive genetic group of 13 varieties from little Kabylie has been identified among the main Algerian olive cultivars and it can be considered a valuable genetic resource for future cultivation and breeding programs. Nuclear and chloroplast genetic profiles provided here will be useful for future program of plant material certification in Algeria. Bayesian and hierarchical cluster analyses allowed to develop inferences on the different patterns of genetic diversity observed. A detailed evolutionary view of Algerian germplasm has been defined, highlighting its genetic relationship with reference cultivars from the whole Mediterranean Basin and Near East. Our findings are compatible with the hypothesis of the existence of an independent olive domestication area in the center of the Mediterranean Basin, but further analysis with more extended datasets are needed to verify this hypothesis. Unfortunately, in contrast to other species, such as grapevine (Vitis International Variety Catalogue, European Vitis Database), there is no international database of olive varieties to use as a reference, and the genetic profiles are not always available. The creation of a public database for olive germplasm would greatly foster the elucidation of the history of domestication for this important crop species.

## Figures and Tables

**Figure 1 genes-11-00303-f001:**
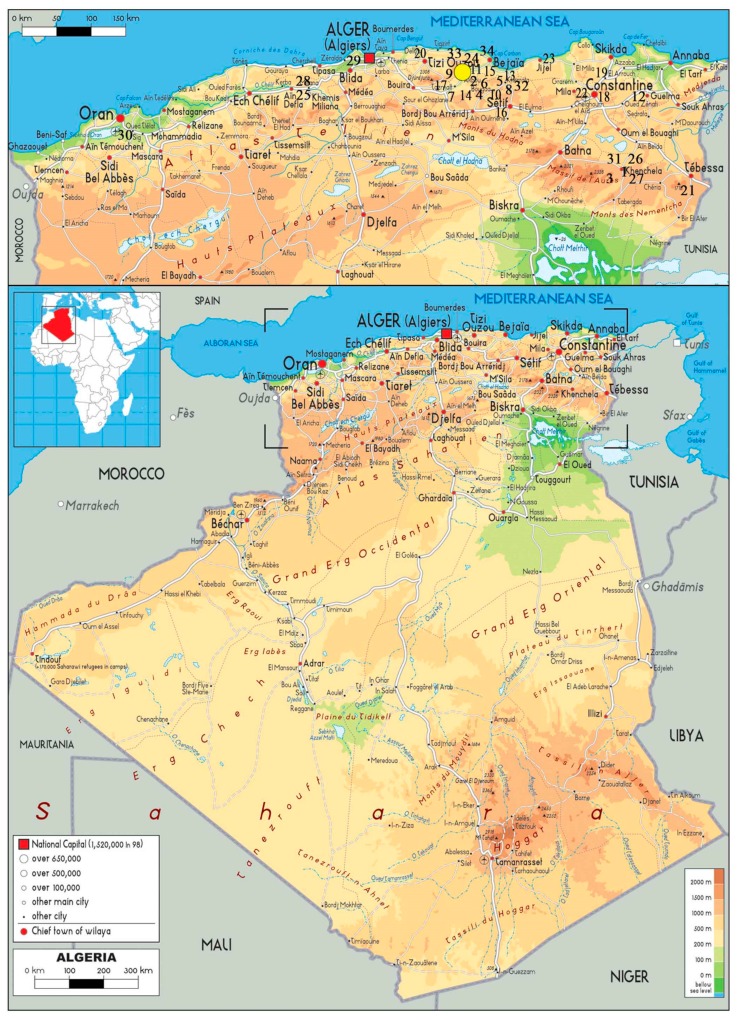
Geographic origin of the Algerian olive cultivars sampled. The yellow point indicates the ITAFV (Takarietz, Bejaia). In brackets, the region of diffusion of the characterized cultivars is indicated. The numbers highlight the origin of cultivars: (1) Abani; (2) Aberkane; (3) Aeleh; (4) Aghchrend’el Ousseur; (5) Aghchren de Titest; (6) Aghenfas; (7) Agrarez; (8) Aguenaou; (9) Aimel; (10) Akerma; (11) Azeradj; (12) Blanquette de Guelma; (13) Bouchouk Guergour; (14) Bouchouk Lafayette; (15) Bouchouk Soummam; (16) Boughenfous; (17) Bouichret; (18) Boukaila; (19) Bouricha; (20) Chemlal; (21) Ferkani; (22) Grosse du Hamma; (23) Hamra; (24) Limli; (25) Longue de Miliana; (26) Mekki; (27) Neb Djemel; (28) Ronde de Miliana; (29) Rougette de Mitidja; (30) Sigoise; (31) Souidi; (32) Tabelout; (33) Takesrit; (34) Tefah.

**Figure 2 genes-11-00303-f002:**
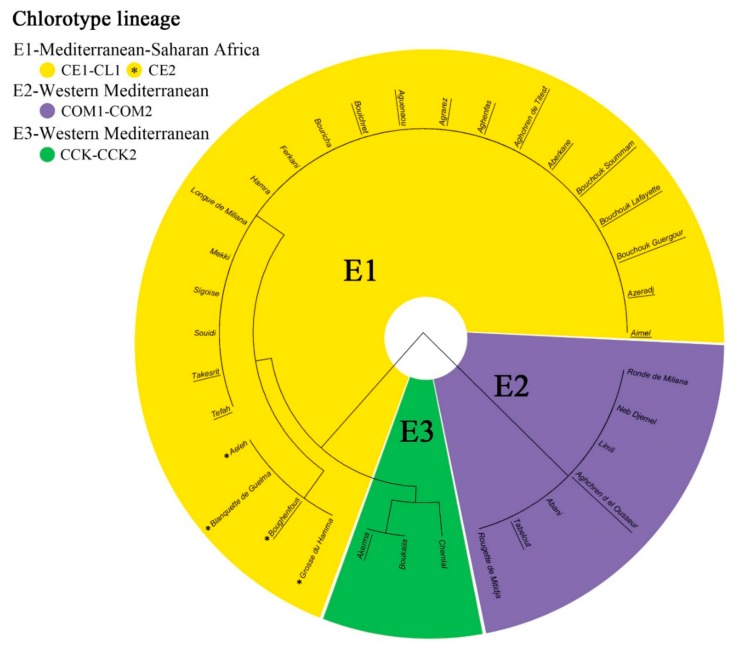
UPGMA (Unweighted Pair-Group Method with Arithmetical Averages) tree of Algerian germplasm based on chloroplast simple sequence repeats (cpSSRs). Original haplotypes (CE1, CE2, COM1-COM2, CCK-CCK2) obtained with cpSSR from Besnard et al. [34] are reported together with the corresponding haplotype lineage: E1, E2 and E3 following Besnard et al. [4].

**Figure 3 genes-11-00303-f003:**
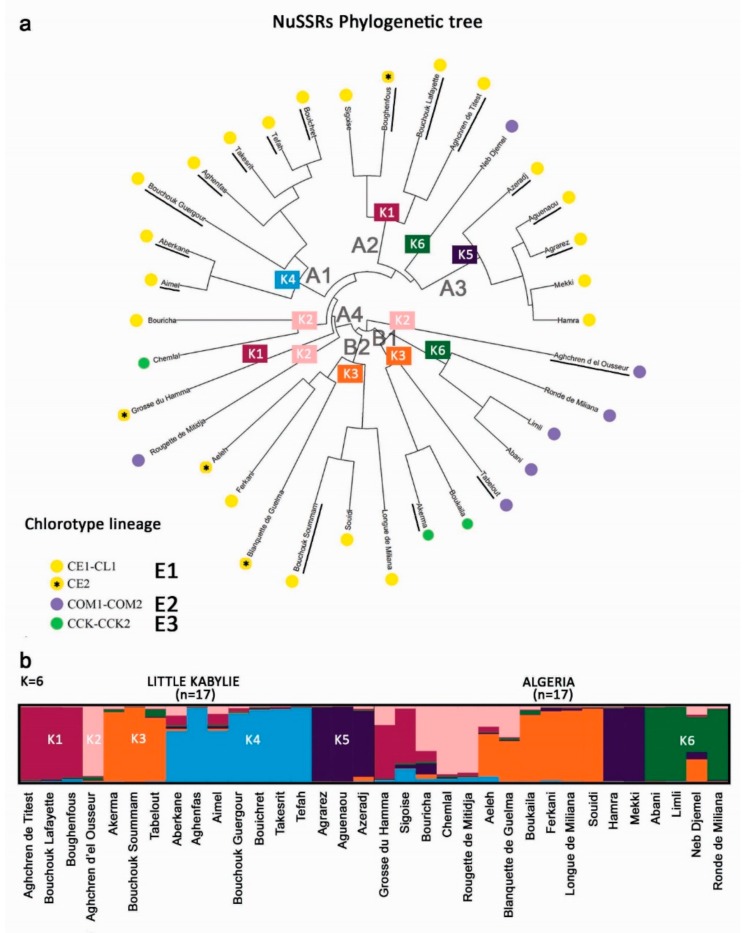
(**a**) UPGMA tree of 34 Algerian varieties based on nuclear simple sequence repeats (nuSSRs). Capital letters indicate the two main clusters (A and B) and relative subclusters; colored dots highlight the haplotype lineage of each variety, and colored rectangles indicate the genetic cluster identified by STRUCTURE analysis. Underlined varieties are native to Little Kabylie (LK). (**b**) STRUCTURE analysis of Algerian germplasm showing on the left side accessions from Little Kabylie and on the right cultivars from other Algerian regions; each color represents the identified genetic cluster and the length of the colored segment shows the estimated membership proportion of each sample to designed group.

**Figure 4 genes-11-00303-f004:**
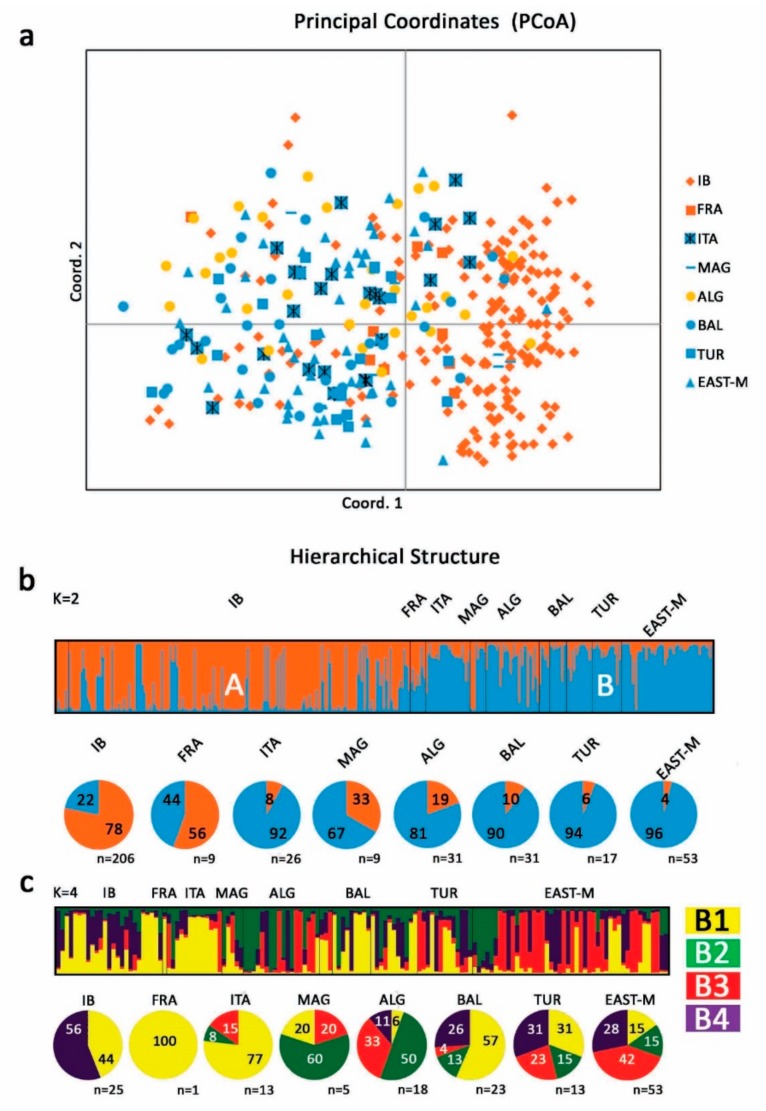
Genetic relationship among Algerian and Mediterranean cultivars. (**a)** Principal Coordinates Analysis (PCoA) and (**b)** STRUCTURE analysis showing unique profiles of 31 Algerian and 351 Mediterranean accessions from Trujillo et al. [40]. (**c**) Hierarchical STRUCTURE analysis of cluster B accessions with ancestry value > 0.80 following Emanuelli et al. [51]. Each color represents the identified genetic cluster (cluster A = orange; cluster B = blue) and the length of the colored segment shows the estimated membership proportion of each sample to the designed group.

**Table 1 genes-11-00303-t001:** List of Algerian varieties analyzed, collected at the Institut Technique de l’Arboriculture Fruitière et de la Vigne (ITAFV).

Accession Number in Figure 1	ID	Cultivar
1	OE-AL-001	Abani
2	OE-AL-002	Aberkane
3	OE-AL-003	Aeleh
4	OE-AL-004	Aghchren d’el Ousseur
5	OE-AL-005	Aghchren de Titest
6	OE-AL-006	Aghenfas
7	OE-AL-007	Agrarez
8	OE-AL-008	Aguenaou
9	OE-AL-009	Aimel
10	OE-AL-010	Akerma
11	OE-AL-011	Azeradj
12	OE-AL-012	Blanquette de Guelma
13	OE-AL-013	Bouchouk Guergour
14	OE-AL-014	Bouchouk Lafayette
15	OE-AL-015	Bouchouk Soummam
16	OE-AL-016	Boughenfous
17	OE-AL-017	Bouichret
18	OE-AL-018	Boukaila
19	OE-AL-019	Bouricha
20	OE-AL-020	Chemlal
21	OE-AL-021	Ferkani
22	OE-AL-022	Grosse du Hamma
23	OE-AL-023	Hamra
24	OE-AL-024	Limli
25	OE-AL-025	Longue de Miliana
26	OE-AL-026	Mekki
27	OE-AL-027	Neb Djemel
28	OE-AL-028	Ronde de Miliana
29	OE-AL-029	Rougette de Mitidja
30	OE-AL-030	Sigoise
31	OE-AL-031	Souidi
32	OE-AL-032	Tabelout
33	OE-AL-033	Takesrit
34	OE-AL-034	Tefah

**Table 2 genes-11-00303-t002:** Comparison of chlorotype lineages of Algerian varieties identified in this study and in published literature.

	Accession Name	Origin	Chlorotype Lineage		
			This Study	Besnard et al. [4]	Houane et al. [18]
1	Abani	OWGB-Marrakech - ITAFV- Takarietz	E2	E2-1	E3
2	Aberkane	OWGB-Marrakech - ITAFV- Takarietz	E1	E1-1	-
3	Aeleh	ITAFV- Takarietz	E1	-	-
4	Aghchren de Titest	OWGB-Marrakech - ITAFV- Takarietz	E1	E1-1	-
5	Aghchren d’Elousseur/Azeradj Tamorka	OWGB-Marrakech - ITAFV- Takarietz	E2	E1-1	-
6	Aghenfas	ITAFV- Takarietz	E1	-	-
7	Agrarez	OWGB-Marrakech - ITAFV- Takarietz	E1	E1-1	-
8	Aguenaou	OWGB-Marrakech - ITAFV- Takarietz	E1	E1-2	-
9	Aharoune	OWGB-Marrakech	-	E3-2	-
10	Ahia Ousbaa	OWGB-Marrakech	-	E2-1	E3
11	Aîmel	OWGB-Marrakech - ITAFV- Takarietz	E1	E1-1	-
12	Akenane	OWGB-Marrakech	-	E1-1	-
13	Akerma	OWGB-Marrakech - ITAFV- Takarietz	E3	E1-1	-
14	Azeboudj de Khirane	OWGB-Marrakech	-	E2-1	E3
15	Azeradj	OWGB-Marrakech - ITAFV- Takarietz	E1	E1-1	-
16	Blanquette de Castu	OWGB-Marrakech	-	E1-1	-
17	Blanquette de Guelma	OWGB-Marrakech - ITAFV- Takarietz	E1	E1-2	-
18	Bouchouk Lafayette	OWGB-Marrakech - ITAFV- Takarietz	E1	E1-1	-
19	Bouchouk Soummam	OWGB-Marrakech - ITAFV- Takarietz	E1	E1-1	-
20	Bouchouk_Guergour	ITAFV- Takarietz	E1	-	-
21	Boughenfous	ITAFV- Takarietz	E1	-	-
22	Bouichret	ITAFV- Takarietz	E1	-	-
23	Boukaïla	OWGB-Marrakech - ITAFV- Takarietz	E3	E1-1	-
24	Bouricha	OWGB-Marrakech - ITAFV- Takarietz	E1	E3-3	-
25	Chemlal de Kabylie	OWGB-Marrakech - ITAFV- Takarietz	E3	E3-2	-
26	Ferkani/Jemri bouchouka	OWGB-Marrakech - ITAFV- Takarietz	E1	E1-1	-
27	Grosse du Hamma	OWGB-Marrakech - ITAFV- Takarietz	E1	E1-2	-
28	Hamra	OWGB-Marrakech - ITAFV- Takarietz	E1	E3-3	-
29	Ifiri	OWGB-Marrakech	-	E1-1	-
30	Khadraïa	OWGB-Marrakech	-	E2-1	E3
31	Limli	OWGB-Marrakech - ITAFV- Takarietz	E2	E1-1	-
32	Longue de Meliana	OWGB-Marrakech - ITAFV- Takarietz	E1	E1-1	-
33	Mekki	OWGB-Marrakech - ITAFV- Takarietz	E1	E1-1	-
34	Neb jmel	OWGB-Marrakech - ITAFV- Takarietz	E2	E2-1	E3
35	Ronde de Meliana	OWGB-Marrakech - ITAFV- Takarietz	E2	E1-1	-
36	Rougette de Metidja	OWGB-Marrakech - ITAFV- Takarietz	E2	E1-1	E3
37	Sigoise	ITAFV- Takarietz	E1	-	-
38	Souidi	OWGB-Marrakech - ITAFV- Takarietz	E1	E2-1	E3
39	Tabelout	ITAFV- Takarietz	E2	-	-
40	Taksrit	OWGB-Marrakech - ITAFV- Takarietz	E1	E1-1	-
41	Tefah	OWGB-Marrakech - ITAFV- Takarietz	E1	E1-1	-
42	Zeboudj Boudoudan	OWGB-Marrakech	-	E1-1	-
43	Zeletni	OWGB-Marrakech	-	E2-1	E3

**Table 3 genes-11-00303-t003:** Genetic parameters of Algerian and Mediterranean germplasm obtained by nuSSR profiles.

*Pop*	*N*	*Na*	*Ne*	*I*	*Ho*	*He*	*F*
ALG	30.7	7.1	3.7	1.4	0.696	0.674	−0.024
	0.2	0.8	0.6	0.1	0.085	0.053	0.083
IB	205.4	11.0	4.1	1.5	0.796	0.684	−0.166
	0.3	2.4	0.8	0.2	0.077	0.066	0.023
FRA	8.7	4.7	3.2	1.2	0.732	0.635	−0.171
	0.2	0.7	0.5	0.2	0.075	0.067	0.071
ITA	25.9	7.4	4.5	1.6	0.857	0.734	−0.169
	0.1	1.3	0.8	0.2	0.057	0.047	0.031
MAG	9.0	5.3	3.5	1.3	0.762	0.660	−0.130
	0.0	0.4	0.5	0.1	0.095	0.068	0.080
BAL	31.0	8.0	4.6	1.6	0.793	0.725	−0.085
	0.0	1.4	0.7	0.2	0.087	0.068	0.037
TUR	17.0	8.3	4.7	1.7	0.756	0.728	−0.045
	0.0	1.2	0.7	0.2	0.076	0.068	0.052
EAST-M	52.6	10.3	5.2	1.8	0.785	0.746	−0.049
	0.3	1.8	1.0	0.2	0.070	0.062	0.030
Mean	47.5	7.8	4.2	1.5	0.772	0.698	−0.105
	8.2	0.5	0.3	0.1	0.027	0.021	0.020

Mean value over loci and standard errors for each population: N: Number of samples; Na: Number of different alleles; Ne: Number of effective alleles; I: Shannon’s information index; He: Expected heterozygosity; Ho: Observed heterozygosity; F: Inbreeding coefficient. ALG: Algeria; IB: Iberian Peninsula—Spain and Portugal; FRA: France; ITA: Italy; MAG: Maghreb—Morocco and Tunisia; BAL: Balcanic Peninsula—Croatia, Albania and Greece; TUR: Turkey—Turkey and Cyprus; EAST-M: East Mediterranean—Iran, Israel, Lebanon, Syria and Egypt.

**Table 4 genes-11-00303-t004:** Estimates of pairwise Gst values (below the diagonal) and Unbiased Nei’s genetic distance (above the diagonal) among overall populations.

	*IB*	*FR*	*ITA*	*MAG*	*ALG*	*BAL*	*TUR*	*EAST-M*
***IB***		0.119	0.159	0.057	0.286	0.179	0.188	0.168
***FR***	**0.024**		0.134	0.139	0.272	0.125	0.073	0.136
***ITA***	**0.030**	**0.024**		0.153	0.242	0.051	0.087	0.119
***MAG***	**0.017**	**0.027**	**0.023**		0.226	0.111	0.137	0.163
***ALG***	**0.051**	**0.045**	**0.040**	**0.044**		0.193	0.261	0.245
***BAL***	**0.033**	**0.022**	**0.008**	**0.013**	**0.032**		0.052	0.094
***TUR***	**0.033**	0.015	**0.013**	0.015	**0.044**	0.009		0.025
***EAST-M***	**0.031**	**0.024**	**0.018**	**0.022**	**0.040**	**0.014**	0.004	

In bold significant values with *p* ≤ 0.01 calculated over 999 permutations.

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
