# Peer review of "Molecular Analysis of the Official Algerian Olive Collection Highlighted a Hotspot of Biodiversity in the Central Mediterranean Basin"

_genes, 2020, doi:10.3390/genes11030303_

Round 1

Reviewer 1 Report

In this study the authors have deeply characterized Algerian olive germplasm through nuclear and chloroplast microsatellites, identifing the presence of a biodiversity hotspot in the Central Mediterranean basin. Interestingly, they also shed some light on the genetic relationship of Algerian and Mediterranean olive germplasm hypothesizing events of secondary domestication and/or cross and hybridization. The issue is certainly of considereble interest given the economic and cultural importance of the olive specie. A perplexity that needs to be clarified is whether authors’ results are actually comparable with data in the different databases they considered, due to the use of different lab staff, sequencers, operators, taq shift, etc. Did the authors create a consensus dataset?

The research is in accordance with the aims of the journal and has been well described.

Moreover, the manuscript is written in a clear English language, although some typing errors and mistakes sometimes occur (see below).

The manuscript needs some minor revisions before pubblication.

Raw 76: ref. 28 and 29 refers to grape and not to olive. Please, check the right pubblication of Sefc (maybe Sefc, K.M., Lopes, M.S., Mendonça, D., Rodrigues Dos Santos, M, Da Câmara Machado, M. & Da Câmara Machado, A. (2000) Identification of microsatellites loci in olive (Olea europaea L.) and their characterisation in Italian and Iberian olive trees. Molecular Ecology, 9, 1171–1193).

Raw 100: please, clarify if authors mean “Geographic origin” or sample site.

Raw 142: remove “were”

Raw 173: please clarify the meaning of “original”

Raw 174: close round backets and remove the extra space

Table S6: please, add the meaning of Ne and F and remove Nu

Raw 176 and 177: authors speak about 14 alleles per DCA09, while in table S6 they report 10 alleles. Consequently, the mean values (8.7 or 7.2?) reported in text and table are discrepant. Please check.

Raw 181: delete dot

Raw 186: remove the extra space

Raw 290: close backets

Raw 384: substitute (Bersnard et al., 2007) with [34].

Author Response

First of all, we thank both reviewers and editor for positive and constructive comments and suggestions. The following is a point-to-point response to reviewer 1 comments, as listed below.

Reviewer 1

In this study the authors have deeply characterized Algerian olive germplasm through nuclear and chloroplast microsatellites, identifing the presence of a biodiversity hotspot in the Central Mediterranean basin. Interestingly, they also shed some light on the genetic relationship of Algerian and Mediterranean olive germplasm hypothesizing events of secondary domestication and/or cross and hybridization. The issue is certainly of considereble interest given the economic and cultural importance of the olive specie.

A perplexity that needs to be clarified is whether authors’ results are actually comparable with data in the different databases they considered, due to the use of different lab staff, sequencers, operators, taq shift, etc. Did the authors create a consensus dataset?

Answer: we thank the reviewer for his/her comments. We must specify that for cpSSRs we did not find any difficulties to compare our results because cpSSRs profiles were always available e.g. Besnard et al., [34] Haouane et al. [18].  We understand the perplexity concerning SSRs data, as before starting our analyses we try to compare our data with different datasets without obtaining reliable results (e.g. http://www.oleadb.it/). Furthermore nuSSRs genetic profiles were not always available: e.g. we asked many times to Dr. Haouane the dataset analysed in Haouane et al.[18] without obtaining any answer. Dataset of Trujillo et al 2014 [40] was the only dataset available that we could use, because we share a common variety with the same genetic profile at 7 SSRs that allowed us to normalize the data, as indicated in M&M raw 124Normalization among datasets was achieved using the common variety Chemlal de Kabylie, present in our dataset with the synonym Chemlal.”

In order to share with the scientific community the results obtained in the present paper for future analyses, we added in the Table S3 different sheets with cpSSR and nuSSRs profiles of Algerian varieties, respectively, and the public Mediterranean dataset (Trujillo et al.[40]) used at the 7 common SSRs

The research is in accordance with the aims of the journal and has been well described.

Moreover, the manuscript is written in a clear English language, although some typing errors and mistakes sometimes occur (see below).

Raw 76: ref. 28 and 29 refers to grape and not to olive.

Answer: we thank the reviewer for his/her comments. Following his/her suggestion, we attributed the references to the correct species Please, check the right pubblication of Sefc (maybe Sefc, K.M., Lopes, M.S., Mendonça, D., Rodrigues Dos Santos, M, Da Câmara Machado, M. & Da Câmara Machado, A. (2000) Identification of microsatellites loci in olive (Olea europaea L.) and their characterisation in Italian and Iberian olive trees. Molecular Ecology, 9, 1171–1193).

Raw 100: please, clarify if authors mean “Geographic origin” or sample site.

Answer: we thank the reviewer for his/her comments. Authors mean sample site, where germplasm was collected before introduction in the official Algerian collection. We modified the text as follows:

” The numbers highlight the sample sites where germplasm was collected before introduction at the ITAFV collection”

Raw 142: remove “were”

Answer: we thank the reviewer for his/her comments and we modified the text accordingly

Raw 173: please clarify the meaning of “original”

Answer: we thank the reviewer for his/her comments and we modified the text following his/her suggestion: “Original haplotypes (CE1, CE2, COM1-COM2, CCK-CCK2) obtained with cpSSRs from Besnard et al. [34] are reported together with the corresponding haplotype lineage: E1, E2 and E3 following Besnard et al. [4].

Raw 174: close round backets and remove the extra space

Answer: we thank the reviewer for his/her comments and we modified the text accordingly

Table S6: please, add the meaning of Ne and F and remove Nu

Answer: we thank the reviewer for his/her comments and we modified the text accordingly.

.

Raw 176 and 177: authors speak about 14 alleles per DCA09, while in table S6 they report 10 alleles. Consequently, the mean values (8.7 or 7.2?) reported in text and table are discrepant. Please check.

Answer: we thank the reviewer for his/her comments. The values of the table S6 are the correct ones, we modified the values in the text.

Raw 181: delete dot

Answer: we thank the reviewer for his/her comments and we modified the text accordingly.

.

Raw 186: remove the extra space

Answer: we thank the reviewer for his/her comments and we modified the text accordingly.

Raw 290: close backets

Answer: we thank the reviewer for his/her comments and we modified the text accordingly.

Raw 384: substitute (Bersnard et al., 2007) with [34].

Answer: we thank the reviewer for his/her comments and we modified the text accordingly.

Reviewer 2 Report

Dear Authors,

The manuscript “Olive biodiversity hotspot in the Central Mediterranean basin: nuclear and chloroplast microsatellites analysis of the official Algerian varieties collection is well written an easy to follow and the data are correctly managed. The manuscript should benefit from English language editing service. Overall, I would suggest correcting/rephrasing some issues in the manuscript, that are added as comments in the attached pdf document.

It was a pleasure to review.

Author Response

First of all, we thank both reviewers and editor for positive and constructive comments and suggestions. The following is a point-to-point response to the  reviewer 2 comments, as listed below.

The manuscript “Olive biodiversity hotspot in the Central Mediterranean basin: nuclear and chloroplast microsatellites analysis of the official Algerian varieties collection” is well written an easy to follow and the data are correctly managed. The manuscript should benefit from English language editing service. Overall, I would suggest correcting/rephrasing some issues in the manuscript, that are added as comments in the attached pdf document.

It was a pleasure to review.

Very similar title: Phylogenetic Relationship Among Wild and Cultivated Grapevine in Sicily: A Hotspot in the Middle of the Mediterranean Basin (https://www.frontiersin.org/articles/10.3389/fpls.2019.01506/full)

Answer: the cited paper describes the situation about grape in the Mediterranean Basin. Our manuscript is focused on olive and the results should clarify the relationship of studied species, covering some lacks of knowledge. Therefore, the two works share the same goals, hence they could have similar titles. Anyway, following the suggestion, we modify the original title (“Olive biodiversity hotspot in the Central Mediterranean basin: nuclear and chloroplast microsatellites analysis of the official Algerian varieties collection”) in “Molecular analysis of the official olive Algerian collection highlighted a hotspot of biodiversity in the Central Mediterranean basin”

Raw 102 (reviewer#2’s pdf )Answer: we thank the reviewer for his/her comments and we modified the text of figure captions 1 following his/her suggestion: “In brackets, the region of diffusion of the characterized cultivars is indicated.”

Raw 129-130 (reviewer#2’s pdf)Answer: we thank the reviewer for his/her comments and we modified the text of figure captions 1 following his/her suggestion: “and some Algerian varieties were grouped in single genetic profiles for the Structure analyses because with 7 SSRs they were not able to differentiate, namely: Aguenau including also Agrarez and Hamra; Aimel including also Aberkane. cpSSRs profiles were also compared with available published dataset [4, 34].”

Raw 167-168 (reviewer#2’s pdf): Table 2

Answer: we thank the reviewer for his/her comments and we modified the table 2 eliminating the last column.

Raw 176-178 (reviewer#2’s pdf)

Answer: we thank the reviewer for his/her comments and we modified the text following his/her suggestion: “with an average of 7.2 alleles per marker (Table S6), which is in agreement with previous studies”

Raw 193-195 (reviewer#2’s pdf)

Answer: we thank the reviewer for his/her comments and we modified the text following his/her suggestion: “The last remaining cultivar….”

Raw 195 (reviewer#2’s pdf)

Answer: we thank the reviewer for his/her comments and we modified the text following his/her suggestion: “….to one of the six identified genetic groups (Figure 3b, Table S4), while the remaining four cultivars…”

Raw 205 (reviewer#2’s pdf)

Answer: we thank the reviewer for his/her comments and we modified the text following his/her suggestion: “UPGMA tree of 34 Algerian varieties based on nuSSRs”

Raw 277 (reviewer#2’s pdf)

Answer: we thank the reviewer for his/her comments but we think that the phrase is correct, it is used in different articles such as Besnard et al [12] or Diez et al [9]: “For thousands of years, olive cultivation has been central in the culture and economy of many Mediterranean and Middle Eastern regions.”

Raw 291 (reviewer#2’s pdf)

Answer: we thank the reviewer for his/her comments and we modified the text following his/her suggestion:“10 of which have never been described”

Raw 300-302 (reviewer#2’s pdf)

Answer: we thank the reviewer for his/her comments and we modified the text as follows: “These discrepancies are probably due to the different panel of varieties analysed and the presence of wild germplasm in previous studies, which contained private alleles [14, 17].”

Raw 304 (reviewer#2’s pdf)

Answer: we thank the reviewer for his/her comments and we modified the text following his/her suggestion:” For example, we found that Agrarez and Azeradj had distinct profiles at 8 loci”

Raw 321-322 (reviewer#2’s pdf)

Answer: we thank the reviewer for his/her comments and we modified the text as follows: Our results mostly confirmed the chlorotypes identified in previous studies (n=18) but highlighted some divergence (Table 2); in particular, we found that for nine varieties chlorotypes were assigned differently as compared to Besnard et al. [4], six when comparing Besnard et al. [4] and Haouane et al. [18], and two when comparing the three datasets.”

Raw 324 (reviewer#2’s pdf)

Answer: we thank the reviewer for his/her comments and we modified the text following his/her suggestion: “mislabelling errors in the WOGB collection of Marrakech”

Raw 346 (reviewer#2’s pdf)

Answer: we thank the reviewer for his/her comments and we modified the text adding the point and changing “unevenly” with “differently”

Raw 357 (reviewer#2’s pdf)

Answer: we thank the reviewer for his/her comments and we modified the text as follows: ”We found that the highest proportion of Algerian varieties”

Raw 361-362 (reviewer#2’s pdf)

Answer: we thank the reviewer for his/her comments and we eliminated the two articles.

Raw 362-363 (reviewer#2’s pdf)

Answer: we thank the reviewer for his/her comments and we modified the text following his/her suggestion: ”in particular, we can speculate that few varieties with chlorotype E2 probably represent locally domesticated cultivars or cultivars imported from a central Mediterranean region such as Italy”

Raw 388 (reviewer#2’s pdf)

Answer: we thank the reviewer for his/her comments, we eliminated the citation.

“domestication center in the Central Mediterranean area [9, 55].”

Raw 398 (reviewer#2’s pdf)

Answer: we thank the reviewer for his/her comment; we modified the text following his/her suggestion:

reconsidered to evaluate its real contribution.”

Raw 399-401 (reviewer#2’s pdf)

Answer: we thank the reviewer for his/her comments but the common name of wild olive tree is correctly written and probably the reviewer made a typographical error as the suggested name “olesaters”, it is not correct. We modified the text as follows: ”We can hypothesize that before foreign civilizations arrived in Algeria, wild olive tree populations consisted of two taxa, oleaster (Olea europaea subsp. sylvestris) and Laperrine’s olive (Olea europaea subsp. laperrinei) that were already exploited by native local human populations, laying the foundations for the development of olive cultivation.

Raw 403 (reviewer#2’s pdf)

Answer: we thank the reviewer for his/her comments, we modified the text following his/her suggestion:”Local people and settlers from abroad”

Raw 416 (reviewer#2’s pdf)

Answer: we thank the reviewer for his/her comments; we modified the text following his/her suggestion: “it can be considered a valuable genetic resource for future cultivation and breeding programs”

Raw 424-426 (reviewer#2’s pdf)

Answer: we thank the reviewer for his/her comments; we modified the text following his/her suggestion: “there is no international database of olive varieties to use as a reference, and the genetic profiles are not always available”